Accepted at the ICLR 2024 Workshop on AI4Differential Equations In Science

# Clifford Neural Operators on Atmospheric Data Influenced Partial Differential Equations

**Sujit Roy\***
Earth System Science Center
The University of Alabama in Huntsville
Huntsville, AL, USA
{sujit.roy}@nasa.gov

**Wei Ji Leong**
Development Seed
Washington, DC, USA
{weiji}@developmentseed.org

**Rajat Shinde, Christopher E. Phillips, Ankur Kumar**
Earth System Science Center
The University of Alabama in Huntsville
Huntsville, USA

**Manil Maskey, Rahul Ramachandran**
NASA Marshall Space Flight Center
Huntsville, AL, USA

## Abstract

Mathematical representations of the atmosphere are key to forecasting and research tasks across Earth science. Numerically solving the underlying partial differential equations (PDEs) of the atmosphere, however, can be difficult and computationally expensive with numerous trade-offs between computing efficiency and accuracy. Utilizing neural networks to learn approximations of the PDE solutions from the data can help us model complex phenomena more efficiently than traditional numerical schemes. Here, we have applied Clifford algebra-based neural operators for predicting atmospheric variables. Clifford Fourier neural operators are used with two different backbone architectures, ResNet and UNet, on custom data of U10, V10, and surface pressure as well as U500, V500, and Z500. Clifford Fourier neural operators, coupled with ResNet and UNet architectures, are applied to a key reanalysis dataset. Model performance is initially strong, but we observe increasing errors, resulting in the model becoming highly unstable.

## 1 Introduction

Mathematical representations and specifically partial differential equations (PDEs) are central to modeling scientific and engineering phenomena, often requiring computationally intensive numerical methods like finite differences for solutions Quarteroni & Valli (2008). However, the high computational demand, especially in understanding and modelling atmosphere dynamics, has led to a growing interest in data-driven machine learning approachesKarniadakis et al. (2021). These methods offer promising solutions for efficiently approximating complex PDE systems in various applications, where traditional numerical solvers are often slow and inefficient Li et al. (2020a; 2021); Brandstetter et al. (2022); Kochkov et al. (2021); Raonić et al. (2023); Pathak et al. (2022); Evans (2022); Lu et al. (2021). Traditional solvers like finite element and finite difference methods balance resolution against speed. Higher resolution increases accuracy but slows computation, especially for complex PDEs requiring fine discretization. In contrast, data-driven methods learn from data, potentially speeding up the process significantly Raissi et al. (2019); Kochkov et al. (2021); Li et al. (2020a). However, standard neural networks, limited to finite-dimensional spaces, are bound to specific discretizations, a drawback for practical use. Thus, the development of mesh-invariant neural networks is essential, as seen in approaches like finite-dimensional operators. To overcome the inherent limitations

of traditional neural networks, there has been the development of an innovative deep-learning framework called *neural operators*. This is fundamentally designed to create direct mappings between function spaces in bounded domains Li et al. (2020a); Raonić et al. (2023); Sergeant-Perthuis et al. (2022); Li et al. (2020b). In this manuscript, we detail the development of a Clifford algebra-based neural operator for the solving of the PDEs that underlie atmospheric dynamics.

## 2 DATASETS

The Modern-Era Retrospective analysis for Research and Applications, Version 2 (MERRA-2) was used for training the new operator(Gelaro et al., 2017). Six variables were selected: U10 (zonal 10 m wind), V10 (meridional 10 m wind), SLP (surface pressure adjusted to sea-level), U500 (500 hPa zonal wind), V500 (500 hPa meridional wind), and Z500 (geopotential height of the 500 hPa level). We selected the data with 3 hours of temporal resolution. For training, we used data from 2021 and for testing, we used 2022. The adjusted pressure was selected to mitigate the influence of terrain on the pressure field. Higher elevation areas naturally have lower pressure that mask the effects of low-pressure weather systems on the wind field. Additionally, at their respective levels (near-surface and 500 hPa), horizontal wind and pressure/geopotential height are strongly coupled by a system of PDEs (Lambaerts et al., 2011). Surface pressure and Z500 both describe the pressure tendencies of the atmosphere, which are crucial to the development and evolution of weather systems, while those systems in turn modify the pressure field (Rostami et al., 2024; Barnes et al., 2022; Jiménez-Esteve & Domeisen, 2022; Spiridonov et al., 2021; Chen et al., 2021; Ning et al., 2021).

The near-surface level was selected due to its immediate relevance to human activities and the 500 hPa layer for its role in large-scale dynamics. The middle atmosphere is less impacted by surface and boundary layer processes such as friction and contains crucial information about vertical transport and steering of synoptic-scale weather systems (Beare, 2007; Jonassen et al., 2020). Forecasters often analyze patterns at 500 hPa to identify the presence of troughs, ridges, and other features that can influence surface weather conditions, illustrating its importance for weather prediction. While the formulation herein is not a complete description of the atmosphere due to a lack of thermodynamic components such as heat and moisture, the selected variables do provide an otherwise complete depiction of key atmospheric processes. Thus, they serve as a useful test bed for developing weather prediction models, with similar being used to test new NWP formulations (Galewsky et al., 2016; Côté & Staniforth, 1990).

## 3 METHODS

When constructing the neural operator framework, we utilized the concept of Clifford algebra over $\mathbb{R}$, the space of all real numbers. The dataset used thus has two vector components, i.e. $U_{10}\&U_{500}$ and $V_{10}\&V_{500}$ and one scalar component i.e. surface pressure (SP) and $Z_{500}$. Two designs are tested, one using $U_{10}$, $V_{10}$, and SLP and the other $U_{500}$, $V_{500}$, and $Z_{500}$, In the first case, we considered two basis vectors $e_1$ and $e_2$ and considered $sp$ as scalar. In the second case, we provided zero as a scalar because the model did not train properly when $Z_{500}$ was used. Further, we performed Clifford Fourier convolution to the dual pair of the function given by $\boldsymbol{f}_0 = f_0(x) + f_{12}(x)i_2$ and $\boldsymbol{f}_1 = f_1(x) + f_2(x)i$. The Clifford Fourier transform ($Cl_{0,2}$) for multivector valued functions $\mathbb{R}^2 \to G^2$ and vectors $x, \xi \in \mathbb{R}^2$ is given by eq 1 Brandstetter et al. (2022).

$$\hat{\boldsymbol{f}}(\xi) = \mathcal{F}\{\boldsymbol{f}\}(\xi) = \frac{1}{2\pi} \int_{\mathbb{R}_2} \boldsymbol{f}(x)e^{-2\pi i_2\langle x,\xi\rangle}dx, \forall \xi \in \mathbb{R}^2 \tag{1}$$

where $f(x)$ and $\hat{\boldsymbol{f}}(\xi)$ represents multivector fields in the spatial and the frequency domain and $i_2 = e_1e_2$.

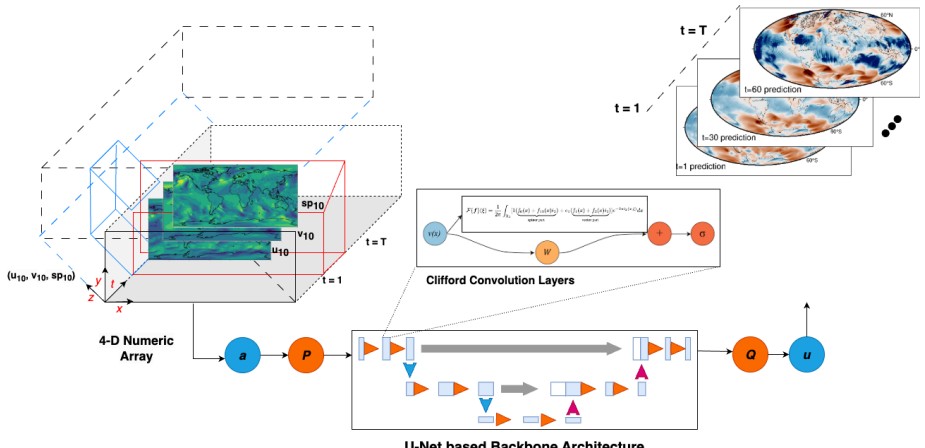

Figure 1: Illustration of the Clifford-based neural operator with U-Net as backbone.

We are going to use the concepts of 4D Clifford algebra $(Cl_{0,2})$ with vector space $G^2$ spanned by $1, e_1, e_2, e_1e_2$. Utilizing the same definition we can rewrite eq 1 as 2.

$$\mathcal{F}\{\boldsymbol{f}\}(\xi) = \frac{1}{2\pi} \int_{\mathbb{R}_2} [1\underbrace{(f_0(x) + f_{12}(x)i_2)}_{\text{spinor part}} + e_1\underbrace{(f_1(x) + f_2(x)i_2)}_{\text{vector part}}]e^{-2\pi i_2\langle x,\xi\rangle}dx$$

$$= 1\left[\mathcal{F}\left(f_0(x) + f_{12}(x)i_2\right)(\xi)\right] + e_1\left[\mathcal{F}\left(f_1(x) + f_2(x)i_2\right)(\xi)\right] \tag{2}$$

We selected two model architectures ResNet-50 He et al. (2016) and U-NetRonneberger et al. (2015) to be used as the backbone for experimentation. Both of the architectures have been used widely in deep learning domains involving images. The convolution layers were replaced with Clifford convolutions Brandstetter et al. (2022); Ruhe et al. (2023), where we used quaternion kernel (g = [-1, -1]). Table 1, shows the architecture of the Clifford convolution with U-Net as backbone. The model was trained by providing a 1-time step input and 1-time step output. We modified the loss $L$ to be mean squared error along with 10 times the change in mean surface pressure $(SP_i)$, as shown in eq 3.

$$L = \frac{1}{n} \sum_{i=1}^{n} \left(Y_i - \hat{Y}_i\right)^2 + 10 * (\frac{1}{N} \sum_{i=1}^{N} SP_i) \tag{3}$$

We used auto-regressive forecasting to create the final evaluation when we provided the model with an initial condition of $t = 0$ and then the output generated by the model as input again.

## 4 RESULTS AND DISCUSSION

Figure 5 shows the prediction, ground truth and difference between prediction and ground truth using U-Net (a) and ResNet (b) architectures respectively for U10, V10 and SLP dataset. We observed in the case of ResNet backbone the model error became excessive after around 30 timesteps. In U-NET, we observed the error to grow large around 60 timesteps ahead. This is not surprising that the U-NET-based backbone was

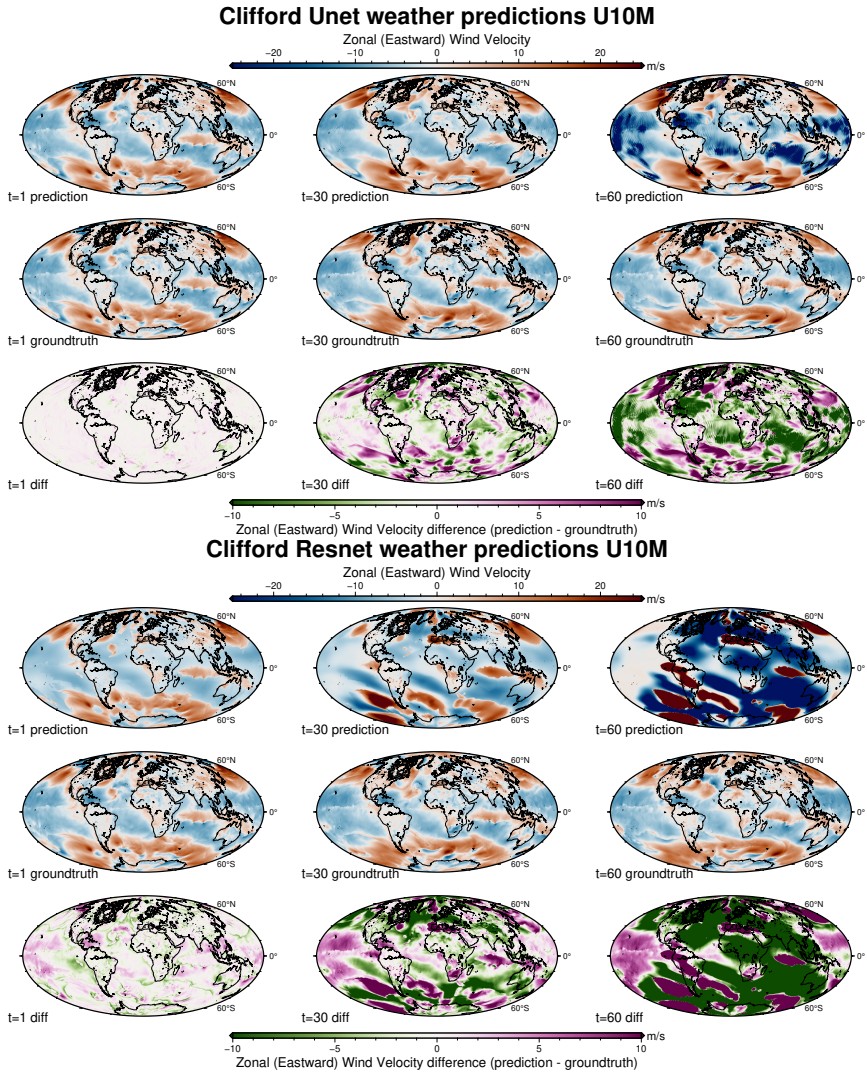

Figure 2: U10 prediction from the initial condition at $t = 0 to 60$ timesteps based on Clifford Fourier convolutions: (a) using U-Net as backbone, (b) using ResNet as the backbone.

more suitable for image mapping. However, other architectures like RNN with SFNO Bonev et al. (2023) have shown longer forecasting with stability.

In figure 5a, there appeared to be two main regions where errors were generated. First, are the mid-to-high latitudes. At timestep 30, regions near the equator had wind speed errors within approximately 3-5 m/s. The exception was near certain regions of activity such as a tropical disturbance in the central Atlantic. At higher latitudes, however, errors approached or surpassed 10 m/s. In these regions, Rosby waves and midlatitude cyclones appeared and were artificially intensified in the model. A potential source of this error was a discrepancy in the height/pressure distribution, the lack of friction near the surface, or the model

not properly learning the relationship between pressure distribution and the wind field. In later timesteps, however, the flow in sub-tropical and tropical regions demonstrated the presence of wavelike features that led to large errors as well. In particular, west of the Americas waves appeared to diffract and experience super-positioning. We think it likely that errors in the pressure field in the tropics produced these fictitious waves. This is because the tropics are barotropic with weak thermodynamic gradients, and thus, disturbances quickly produce gravity waves that transmit their energy to their surroundings. If the AI forecasts a fictitious region of high or low pressure, that feature would produce these waves, which in turn would add energy to the model atmosphere. Consequently, this results in an exponential increase in errors in the tropics. In future, we would like to add more variables for the complexity. Additionally, we would use more training samples and do a longer autoregressive loss.

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

## A APPENDIX

Table 1: Architecture of Clifford based neural operator with U-Net as backbone

| Layer (type:depth-idx) | Input Shape | Output Shape | Param # |
|---|---|---|---|
| CliffordLightningModel | [8, 122, 64, 128, 3] | [8, 122, 64, 128, 3] | – |
| **CliffordG3UNet2d: 1-1** | | | |
| CliffordG3Conv2d: 2-1 | [8, 122, 64, 128, 3] | [8, 32, 64, 128, 3] | 210,816 |
| ModuleList: 2-2 | – | – | – |
| CliffordG3DownBlock: 3-1 | [8, 32, 64, 128, 3] | [8, 32, 64, 128, 3] | 111,040 |
| CliffordG3DownBlock: 3-2 | [8, 32, 64, 128, 3] | [8, 32, 64, 128, 3] | 111,040 |
| CliffordG3Downsample: 3-3 | [8, 32, 64, 128, 3] | [8, 32, 32, 64, 3] | 55,296 |
| CliffordG3DownBlock: 3-4 | [8, 32, 32, 64, 3] | [8, 64, 32, 64, 3] | 344,832 |
| CliffordG3DownBlock: 3-5 | [8, 64, 32, 64, 3] | [8, 64, 32, 64, 3] | 443,264 |
| CliffordG3Downsample: 3-6 | [8, 64, 32, 64, 3] | [8, 64, 16, 32, 3] | 221,184 |
| CliffordG3DownBlock: 3-7 | [8, 64, 16, 32, 3] | [8, 128, 16, 32, 3] | 1,377,792 |
| CliffordG3DownBlock: 3-8 | [8, 128, 16, 32, 3] | [8, 128, 16, 32, 3] | 1,771,264 |
| CliffordG3Downsample: 3-9 | [8, 128, 16, 32, 3] | [8, 128, 8, 16, 3] | 884,736 |
| CliffordG3DownBlock: 3-10 | [8, 128, 8, 16, 3] | [8, 256, 8, 16, 3] | 5,508,096 |
| CliffordG3DownBlock: 3-11 | [8, 256, 8, 16, 3] | [8, 256, 8, 16, 3] | 7,081,472 |
| CliffordG3MiddleBlock: 2-3 | [8, 256, 8, 16, 3] | [8, 256, 8, 16, 3] | – |
| CliffordG3BasicBlock2d: 3-12 | [8, 256, 8, 16, 3] | [8, 256, 8, 16, 3] | 7,081,472 |
| CliffordG3BasicBlock2d: 3-13 | [8, 256, 8, 16, 3] | [8, 256, 8, 16, 3] | 7,081,472 |
| ModuleList: 2-4 | – | – | – |
| CliffordG3UpBlock: 3-14 | [8, 512, 8, 16, 3] | [8, 256, 8, 16, 3] | 11,407,872 |
| CliffordG3UpBlock: 3-15 | [8, 512, 8, 16, 3] | [8, 256, 8, 16, 3] | 11,407,872 |
| CliffordG3UpBlock: 3-16 | [8, 384, 8, 16, 3] | [8, 128, 8, 16, 3] | 3,836,672 |
| CliffordUpsample: 3-17 | [8, 128, 8, 16, 3] | [8, 128, 16, 32, 3] | 1,572,864 |
| CliffordG3UpBlock: 3-18 | [8, 256, 16, 32, 3] | [8, 128, 16, 32, 3] | 2,853,120 |
| CliffordG3UpBlock: 3-19 | [8, 256, 16, 32, 3] | [8, 128, 16, 32, 3] | 2,853,120 |
| CliffordG3UpBlock: 3-20 | [8, 192, 16, 32, 3] | [8, 64, 16, 32, 3] | 959,872 |
| CliffordUpsample: 3-21 | [8, 64, 16, 32, 3] | [8, 64, 32, 64, 3] | 393,216 |
| CliffordG3UpBlock: 3-22 | [8, 128, 32, 64, 3] | [8, 64, 32, 64, 3] | 713,856 |
| CliffordG3UpBlock: 3-23 | [8, 128, 32, 64, 3] | [8, 64, 32, 64, 3] | 713,856 |
| CliffordG3UpBlock: 3-24 | [8, 96, 32, 64, 3] | [8, 32, 32, 64, 3] | 240,320 |
| CliffordUpsample: 3-25 | [8, 32, 32, 64, 3] | [8, 32, 64, 128, 3] | 98,304 |
| CliffordG3UpBlock: 3-26 | [8, 64, 64, 128, 3] | [8, 32, 64, 128, 3] | 178,752 |
| CliffordG3UpBlock: 3-27 | [8, 64, 64, 128, 3] | [8, 32, 64, 128, 3] | 178,752 |
| CliffordG3UpBlock: 3-28 | [8, 64, 64, 128, 3] | [8, 32, 64, 128, 3] | 178,752 |
| Identity: 2-5 | [8, 32, 64, 128, 3] | [8, 32, 64, 128, 3] | – |
| CliffordG3LinearVSiLU: 2-6 | [8, 32, 64, 128, 3] | [8, 32, 64, 128, 3] | – |
| Conv3d: 3-29 | [8, 32, 64, 128, 3] | [8, 32, 64, 128, 1] | 128 |
| CliffordG3Conv2d: 2-7 | [8, 32, 64, 128, 3] | [8, 122, 64, 128, 3] | 210,816 |

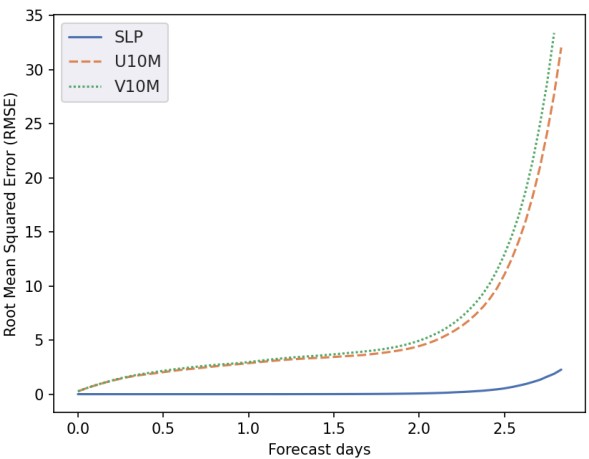

Figure 3: RMSE of U10M, V10M and SLP prediction from the initial condition at t=0 to 2.5 days based on Clifford Fourier convolutions with U-Net as backbone

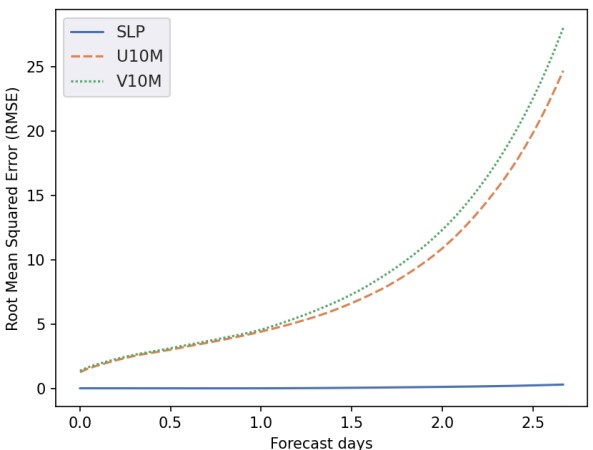

Figure 4: RMSE of U10M, V10M and SLP prediction from the initial condition at t=0 to 2.5 days based on Clifford Fourier convolutions with ResNet as backbone

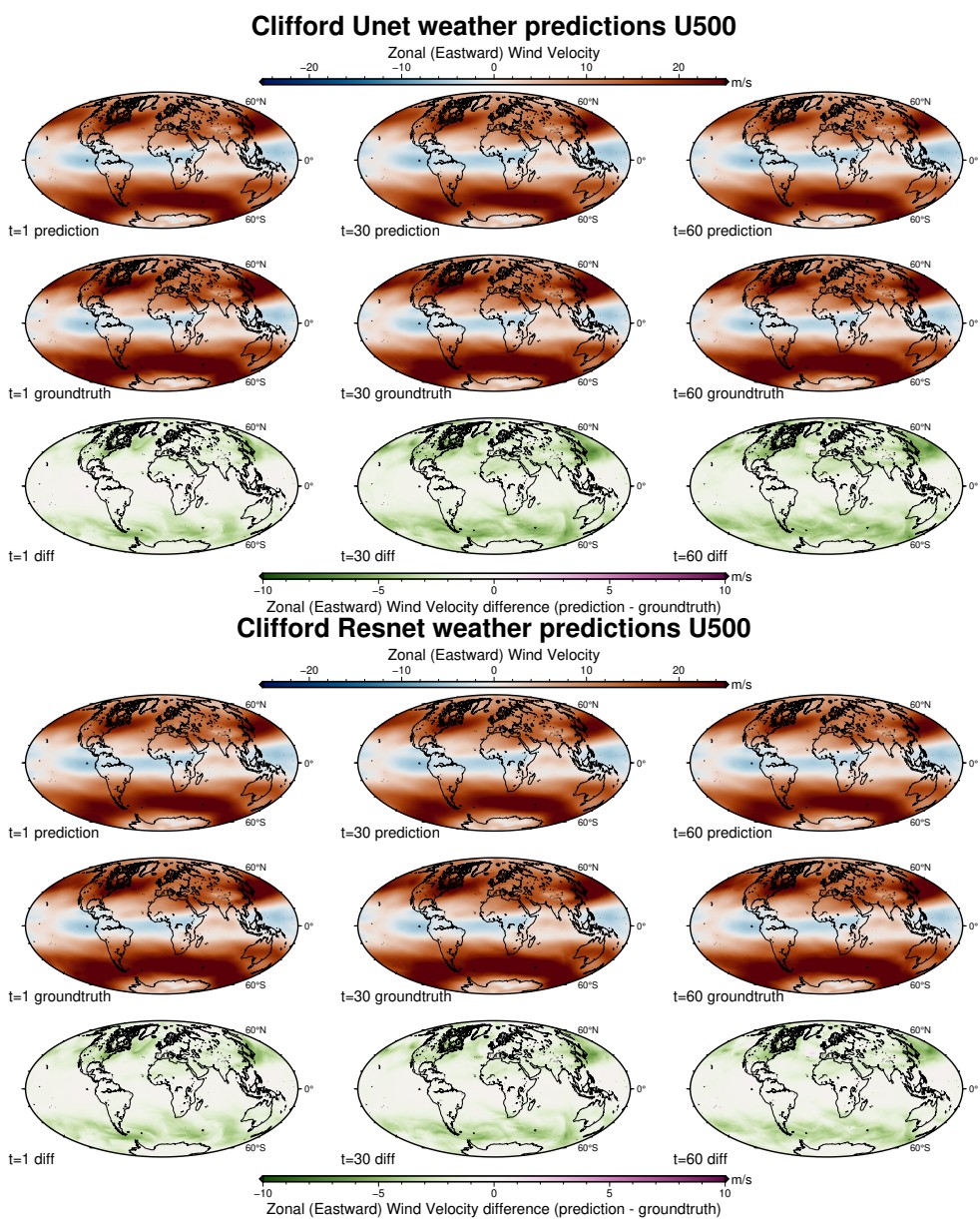

Figure 5: U500 prediction from the initial condition at t=0 to 60 timesteps based on Clifford Fourier convolutions: (a) using U-Net as backbone, (b) using ResNet as the backbone.

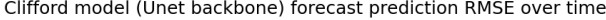

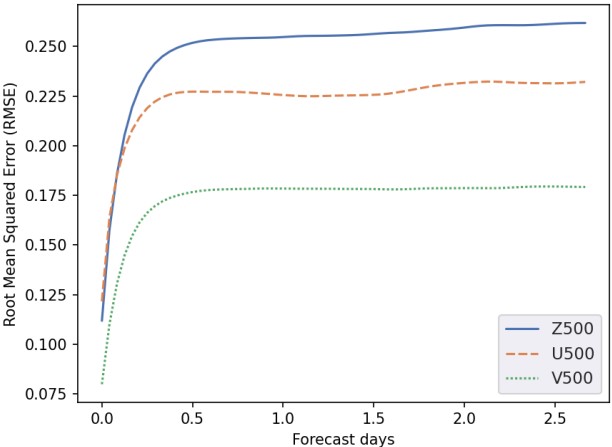

Figure 6: RMSE of Z500, U500 and V500 prediction from the initial condition at t=0 to 2.5 days based on Clifford Fourier convolutions with U-Net as backbone

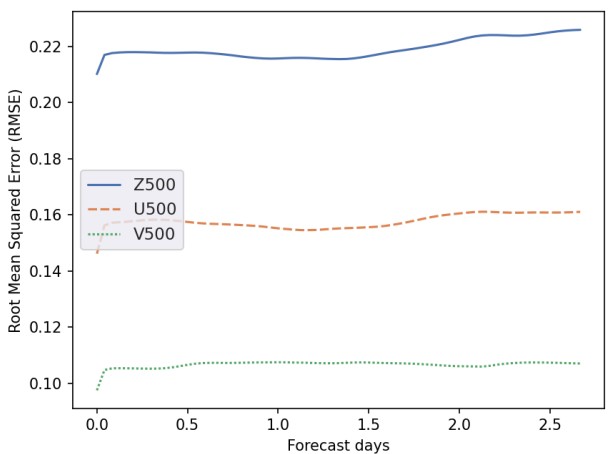

Figure 7: RMSE of Z500, U500 and V500 prediction from the initial condition at t=0 to 2.5 days based on Clifford Fourier convolutions with ResNet as backbone

