# OpenReview forum: "CLIFFORD NEURAL OPERATORS ON ATMOSPHERIC DATA INFLUENCED PARTIAL DIFFERENTIAL EQUATIONS"
_ICLR.cc/2024/Workshop/AI4DiffEqtnsInSci — AI4DiffEqtnsInSci @ ICLR 2024 Poster_

### Official Review · Reviewer_4vLb · 2024-02-15
**CLIFFORD NEURAL OPERATORS ON ATMOSPHERIC DATA INFLUENCED PARTIAL DIFFERENTIAL EQUATIONS**

**Rating:** 6
**Confidence:** 4

**Review:**

This work presents an innovative approach to atmospheric modeling by leveraging Clifford algebra-based neural operators within ResNet and UNet architectures. The application of these methods to predict atmospheric variables, including U10, V10, surface pressure, U500, V500, and Z500, from a key reanalysis dataset, is a significant contribution to the field of Earth science. The use of neural networks to approximate the solutions of partial differential equations (PDEs) is a promising direction, potentially offering a more computationally efficient alternative to traditional numerical schemes.
However, the issue of increasing errors and resultant model instability author had observed warrants careful examination. This instability could be attributed to several factors inherent in the modeling approach or data handling processes.

In conclusion, this study represents a meaningful step forward in the application of advanced neural network architectures for atmospheric modeling. Addressing the observed instability requires a multifaceted approach, examining data preprocessing, model architecture, regularization, and validation strategies. Further research in this direction could significantly enhance our capability to model complex atmospheric phenomena efficiently and accurately.

---

### Official Review · Reviewer_Bpag · 2024-02-24
**This paper explores incorporating Clifford neural operators for Earth forecasting task. The proposed method and its application are novel. However, the empirical studies are not enough to demonstrate its advantages.**

**Rating:** 5
**Confidence:** 4

**Review:**

This paper explores incorporating Clifford neural operators into two types of backbones ResNet and UNet, for Earth forecasting task on MERRA-2 dataset.

Pros
1. The visualization on model design is clear and easy to follow.
2. The visualization on performance comparison helps indentify the areas where the model fail to predict accurately and help analyze the potential causes.

Weaknesses
1. Some expressions are not clear enough. E.g., there lacks descriptions on the notations in Eq(1)(2).
2. The performance drops significantly when the forecasting horizon gets longer. Besides the potential causes discussed in Section 4, the single-input-single-output formulation is also a possible cause that leads to distribution mismatch and accumulated errors.
3. The major contribution is the usage of Clifford neural operator and its application to Earth forecasting. However, there is no evidence that it really results in performance gain. Ablation studies on Clifford neural operator, e.g., comparisons to the original ResNet and UNet, could be helpful.

---

### Meta-Review · Area_Chair_frtu · 2024-03-01

**Recommendation:** Accept (Poster)

**Metareview:**

Both reviewers unanimously agree on acceptance. However, I strongly command the author to address those concerns in the camera ready version.

---

### Decision · Program_Chairs · 2024-03-01

Accept (Poster)